# Modernizing Medical Research to Benefit People and Animals

**DOI:** 10.3390/ani12091173

**Published:** 2022-05-03

**Authors:** Isobel Hutchinson, Carla Owen, Jarrod Bailey

**Affiliations:** Animal Free Research UK, 27 Old Gloucester Street, London WC1N 3AX, UK; carla@animalfreeresearchuk.org (C.O.); jarrod@animalfreeresearchuk.org (J.B.)

**Keywords:** animal research, animal testing, alternative methods, animal replacement, New Approach Methodologies (NAMs)

## Abstract

**Simple Summary:**

The use of animals in research has been the subject of significant public debate and concern. Some argue that animal experiments are essential to medical progress, while others believe that this practice is unethical and does not produce results that can be reliably translated to people. A growing range of modern techniques can replace the use of animals and provide results that are more relevant to human patients. These include the use of human cells and tissues, and computer-based methods. Our work aimed to explore the societal benefits of accelerating the replacement of animals with these new research methods. We found that this approach would benefit animal welfare, public health and the economy. We also found that the British public is generally in favor of efforts to replace animals, and that focusing on this area would help to support the British Government’s policy and economic goals. We suggest that the British Government could greatly expedite the replacement of animal experiments by appointing a new minister to take responsibility for this transition.

**Abstract:**

In the context of widespread public and political concern around the use of animals in research, we sought to examine the scientific, ethical and economic arguments around the replacement of animals with New Approach Methodologies (NAMs) and to situate this within a regulatory context. We also analyzed the extent to which animal replacement aligns with British public and policymakers’ priorities and explored global progress towards this outcome. The global context is especially relevant given the international nature of regulatory guidance on the safety testing of new medicines. We used a range of evidence to analyze this area, including scientific papers; expert economic analysis; public opinion polls and the Hansard of the UK Parliament. We found evidence indicating that replacing animals with NAMs would benefit animal welfare, public health and the economy. The majority of the British public is in favor of efforts to replace animals and focusing on this area would help to support the British Government’s current policy priorities. We believe that this evidence underlines the need for strong action from policymakers to accelerate the transition from animal experiments to NAMs. The specific measure we suggest is to introduce a new ministerial position to coordinate and accelerate the replacement of animals with NAMs.

## 1. Introduction

We believe that there is an urgent need to modernize medical research. We use this term to refer to the process of replacing animal experiments with techniques that have greater direct relevance to human patients. These approaches include the use of advanced cultures of human cells and tissues, artificial intelligence and organ-on-a-chip technology. There is a strong and multi-faceted case for this process of modernization. It is likely to accelerate much-needed medical progress, and would prevent the suffering caused to animals by breeding them, housing them in laboratories and experimenting on them. Within Britain, this transformation would help the Government achieve its current policy priorities and support economic growth. With demonstrable progress taking place around the world, the British Government has much work to do to ensure that Britain is not left behind. Here, we outline the key arguments in favor of modernizing medical research in this way, and recommend decisive policy action that the British Government should take to support this process.

## 2. The Scientific Case for Modernizing Medical Research

The current system of medical research, which relies heavily on animal use, is failing to deliver sufficient progress for patients. Several thousand diseases affect humans, but only around 500 have treatments available [1]. One example is Alzheimer’s disease, which has seen clinical trial failures of more than 99% [2]. A 2019 review concluded that it was not possible to recommend any animal ‘model’ that could provide a reliable prediction of the clinical efficacy of potential treatments for Alzheimer’s [2]. One of the key problems with using animals in disease research is that they often do not naturally suffer from the illness that is being investigated. This results in efforts to artificially induce some symptoms of the disease, whether by physical means or genetic modification. Myriad types of GM animals have been created to investigate general biology, gene functions etc., and also to generate models of human diseases, including multiple infectious diseases such as HIV, immune system defects, blood and metabolic disorders, muscular dystrophy, cancer immunotherapies, and many others [3].

However, genetic modification does little to address the significant differences in genetic complement and gene expression between different species. These, and other, biological differences between species contribute to the high rate of failure in terms of translation to clinical benefit from biomedical research and product safety testing. For example, over 92% of drugs that show promise in animal tests fail to reach the clinic and benefit patients [4]. This high rate of attrition is mainly caused by problems with efficacy and safety that were not predicted by the animal tests.

Modernizing our approach to medical research can not only help to address this extremely poor translation of research to human benefit, but is arguably, given the human ethical issues resulting from such poor translation, absolutely essential. New Approach Methodologies (NAMs) have been defined as ‘new scientific approaches that focus on human biological processes to investigate disease and potential treatments, using human cells, tissues, organs and existing data’ [5].They include the use of advanced cultures of human cells and tissues, artificial intelligence and organ-on-a-chip technology. Since they are often based specifically on humans, and are therefore not hampered by extrapolating from one species to another, these methods provide results that have much greater relevance to patients. NAMs have proved valuable within the context of the COVID-19 pandemic. For example, Benevolent AI used computational tools to identify existing drugs that could be repurposed to treat COVID-19. This process selected a drug called baricitinib, which had anti-viral and anti-inflammatory properties. It is now authorized for the treatment of patients hospitalized with COVID-19 in the USA, Japan and India [6]. Computer-based analyses and algorithms are also being used to examine changes in gene expression caused by COVID-19, and by existing drugs, so that they can be matched—enabling the repurposing of drugs to restore healthy/normal gene expression in patients with COVID-19 [7].

## 3. The Ethical Case for Modernizing Medical Research

Separating the scientific and ethical cases for modernizing medical research may now be an artificial distinction. The ability of medical research to benefit patients is, of course, an ethical question, and so animal research involves human, as well as animal, ethical considerations. Governments and other organizations that use public funds to finance medical research therefore have an ethical duty towards humans to support methods which are most likely to lead to progress. In the field of safety testing, scientists and regulatory bodies should use methods that offer the most accurate predictions, to minimize the risk of harm to people. However, the section below focuses on the more direct ethical issues around using animals in experiments.

It has been argued that the practice of animal research involves an inherent contradiction. It relies on the belief that animals are similar enough to humans to make the results relevant, but that they are different enough to make it morally acceptable to harm them in the pursuit of human benefit. In fact, there is strong evidence to show the opposite: that animals are different in all the ways that make extrapolation of data to humans unreliable, but are so similar in their capacity to experience suffering that this makes a formidable ethical case to not do animal experiments at all, regardless of any proposed human benefit. The Animals (Scientific Procedures) Act 1986 (ASPA), which governs animal experiments in Britain, seeks to minimize suffering as much as possible. However, this idea of humaneness is difficult to reconcile with the reality of experimenting on animals. In his response to a recent debate in the UK Parliament about the use of animals in research, the Minister for Science, Research and Innovation, George Freeman MP stated that ‘the issue of force feeding—which is a controversial term—was raised. I have checked the reason for that, and it is about making sure that the correct dose is administered, but again, the point is well made: we need to make sure that is being done in the most humane and sentient-friendly way’ [8]. There is an obvious conflict between the principle of treating animals humanely and the concept of force-feeding them. As well as the ethical problems with the practice of force-feeding, it also has implications for the results of the experiments, since it causes stress and slows down gastrointestinal function [9].

Before they undergo any experiments, being in a laboratory environment is likely to cause suffering to animals. As discussed in a 2017 editorial by Dr Jarrod Bailey, factors that contribute to the stress and distress of animals in laboratories include a lack of control over the ‘procedures’ that they are subjected to, being prevented from expressing many natural behaviors and unnatural light and noise conditions. This has significant adverse consequences for both science and animal welfare [10]. Within the European Union, animal experiments are governed by Directive 2010/63/EU, which divides experiments into four categories of severity: non-recovery, mild, moderate and severe. The examples provided within Annexe VIII of the Directive illustrate the suffering that can be endured. Experiments that are likely to be classified as ‘mild’ include short-term restraint in metabolic cages or solitary caging of some animals. ‘Moderate’ experiments could be withdrawing food for 48 h in rats or evoking avoidance and escape reactions where the animal cannot avoid the cause of the stimulus. ‘Severe’ experiments include inescapable electric shock to produce learned helplessness or forced swim tests with exhaustion as the end-point. Britain has maintained these severity categories, along with the main substance of the Directive, post-Brexit. We would suggest that these examples are unlikely to reflect the lay-person’s understanding of the terms ‘mild’, ‘moderate’ and ‘severe’, since even ‘mild’ experiments can involve significant suffering. It should be remembered that legislation on animal experiments, such as ASPA, serves to facilitate actions that would often constitute illegal animal cruelty if performed by an unlicensed member of the public in normal circumstances.

In Britain, animal experiments are regulated by the Animals in Science Regulation Unit (ASRU), within the Home Office. Beyond the harms that are legally sanctioned, ASRU’s own reports detail additional suffering caused by incidents of neglect. Failures described in the latest edition of the report include mice being left in a cage in an isolator without appropriate food and water for over two weeks. In another incident, mice were left in a laminar flow hood without access to food and water over a weekend. A total of 93% of the incidents detailed in this document were self-reported to the regulator. Of the remaining two incidents, one was discovered during an inspection, and another reported by a whistleblower [11]. There is no way of knowing how many further incidents may have taken place without being reported.

Alongside these specific incidents, we believe that the provisions of ASPA are routinely being undermined. The law makes it clear that animal experiments should only be licensed when no non-animal method is available. However, there is significant evidence to suggest that this important legal requirement is not being adequately enforced. Analysis by Cruelty Free International found that in 2020 in the UK, 452 skin sensitization tests were carried out on mice, even though a validated non-animal method is available [12]. In addition, analysis by Thomas Hartung et al. found evidence of animal tests for ingredients solely used in cosmetic products, even after the EU-wide ban on this practice [13]. A Parliamentary Question by Alex Sobel MP revealed that no applications to conduct animal experiments were refused permission in 2020 [14]. In the European Union, it is estimated that up to three-quarters of a million animal tests take place every year in spite of validated alternatives that have existed for some time [15].

Understanding the process behind license applications makes this less surprising. By the time a researcher submits a license application to ASRU, it is likely that funding for the project will have been secured, and it will have received support from the Animal Welfare Ethical Review Body (AWERB) at the establishment where the work is to take place. While researchers are supposed to thoroughly consider non-animal methods, there is a significant lack of knowledge within the scientific community about these. The AWERBs are not likely to have detailed knowledge of non-animal methods within each specialist research area. This issue is further highlighted by Judith van Luijk’s work surveying animal welfare officers in the Netherlands, whose role includes advising on the 3Rs. While all 15 respondents believed they contributed to implementing refinement, only one believed they contributed to replacement. Over half of the respondents felt that opportunities to implement the 3Rs were being missed [16]. Within the UK, the ASRU inspectors who grant the licenses have limited knowledge and little time. In 2018 (the latest figures available) there were just 22 ASRU inspectors (20.8 full time equivalents) [11]. Analysis of the non-technical summaries of licenses granted during the first half of 2020 showed that researchers often failed to engage meaningfully with the question that asked them to explain their strategy in searching for non-animal alternatives. Analysis by Animal Free Research UK identified 34 examples of a clear lack of engagement. In the most extreme case, a one-word answer was provided, and there were many examples of researchers simply listing a few non-animal methods and stating that these were not suitable, rather than explaining the in-depth research that should have been carried out to identify non-animal approaches. Researchers should be provided with practical training in identifying non-animal methods. One example of this is the free e-learning module that has recently been made available by the Education and Training Platform for Laboratory Animal Science [17].

It is also important to consider how animal experiments affect the researchers who carry them out. A survey amongst people carrying out animal experiments in Poland provided preliminary data on this subject. A total of 72% of the respondents felt an emotional burden as a result of carrying out animal experiments and 63% felt stress. The study’s authors suggest one contributing factor may be the need to closely observe animals and therefore notice the distress they are experiencing, combined with an obligation to continue with the experiments [18]. The risk of negative mental health impacts on researchers who carry out animal experiments, as well as the risk of compassion fatigue, should be seriously considered and presents yet another ethical issue with this practice.

## 4. The Economic Case for Modernizing Medical Research

While the following discussion focuses on the economic case for a more modern approach to research, this also has important ethical dimensions. As the world emerges from the COVID-19 pandemic, policy makers have a duty to facilitate economic recovery, whilst ensuring that public health receives adequate funding.

Firstly, the economic cost of current public health challenges should be considered. While many factors exacerbate these challenges, the current reliance on animal research and slow progress that flows from this should be considered amongst these. A 2014 report produced for Alzheimer’s Research UK estimated that dementia costs the UK economy £23.6 billion, which was predicted to rise to £59.4 billion by 2050. The report also examined how the economic impact would change if new treatments were found. If an intervention was produced that delayed the onset of dementia by five years, the cost to the economy would reduce by 36% (saving over £21 billion that year). However, it is disappointing to note that this 2014 report modeled its predictions on a new intervention being available by 2020 [19]. Clearly, effective new treatments have not yet materialized, and the difficulty in extrapolating from animal experiments to the clinic is likely to be one of the factors that have contributed to this ongoing failure.

The high attrition rate of drugs could also play a role in increasing the cost of providing public healthcare. While many factors contribute to the elevated cost of medicines, high attrition rates have been amongst the causes cited by the pharmaceutical industry, although this explanation has been questioned by others [20]. Analysis by Deloitte identified a significant increase between 2020–2021 on the estimated return on investment that biopharmaceutical companies could expect from their late-stage drug pipelines, but this still only reached an internal rate of return of seven per cent [21]. More accurate models, that give a better prediction of whether the drug will be safe and effective for humans, could help to reduce the current rates of drug failure, and could potentially have a positive impact in lowering the cost of these new medicines. Ultimately, this could translate into savings for public healthcare providers, such as the UK’s National Health Service (NHS), which purchases the drugs.

As well as reducing the cost of public healthcare, human relevant research methods represent an area where Britain could play a leading role in an emerging industry, which could contribute to economic growth. A report commissioned by Animal Free Research UK and produced by the Centre for Economics and Business Research (Cebr) found that the UK NAMs sector is already making a valuable contribution to the British economy and this is likely to grow significantly over the coming years. In 2019, the NAMs sector contributed an estimated £2.3 billion in turnover and £592 million in gross value added (GVA). The report forecasts that by 2026, the NAMs industry will contribute £2.5 billion to UK GDP – an increase of 700% since 2017 [22]. It is worth noting that these predictions did not take account of any future policy interventions. If the British Government were to support this industry, such as by providing specific grant funding or increased advantage within the Research and Development (R&D) tax relief system, the potential for growth would increase further. The economic benefits of NAMs should be considered within the context of the issue of waste in current research. A 2014 series in the *Lancet* considered the issue of research waste. Despite major global investment in biomedical research (US$240 billion in 2010) it acknowledges that a great deal of this fails to produce achievements that benefit people. While it concedes that some waste is inevitable, it suggests that significant changes need to be made to tackle avoidable waste [23]. Currently, very little funding is provided for non-animal research. A 2014 survey found that this was between 0-0.036 per cent of national science R&D expenditure amongst responding EU member states [24].

## 5. Public Support for Modernizing Medical Research

Various opinion polls have suggested a high level of public concern around the issue of animal experiments. A February 2021 YouGov poll, which was commissioned by Animal Free Research UK, found that 60% of respondents would support the British Government providing more financial support for developing alternatives to experimenting on animals in medical research; 68% of respondents would support a policy ending animal experiments in medical research in the UK and replacing them with non-animal alternatives; and 70% of respondents would support animal experiments in medical research being phased out by 2040 [25]. Similarly, a YouGov poll commissioned by Cruelty Free International in September 2021 found that 65% of respondents wanted to see a binding plan in place for ending animal testing [26].

While both these polls were commissioned by organizations that seek to replace the use of animals in research and testing, it should be noted that YouGov is independent of these and has stringent standards on the quality of its research. In addition, the results of these polls are in line with the government-commissioned 2018 Ipsos MORI poll in which three quarters of respondents thought that more should be achieved in finding alternatives to animal experiments [27].

This concern around animal experiments reflects the British public’s wider concerns around animal welfare. In a recent debate on animal testing in Parliament, a number of MPs commented on the level of concern amongst their constituents about animal welfare issues [8]. The current interest in veganism could lead to more people recognizing the rights of animals and questioning whether it is ethical to harm them in the pursuit of human benefit. This focus on animals’ fundamental rights clearly differs from the approach of animal welfare, which accepts the use of animals but seeks to ensure that their suffering is minimized. The number of vegans in Britain quadrupled between 2014-19 [28]. In a 2014 study, over 89% of respondents mentioned issues relating to animals as one of the factors that motivated them in adopting a vegan diet [29].

Animal protection groups often emphasize the millions of animals who are used each year in research—some 200 million globally [30]. While it is vital to highlight the scale of animal use, it may be worth considering how this might impact on public and political concern for animals in laboratories. ‘Compassion fade’ is a well-documented phenomenon, which describes the tendency for compassion to decrease as the number of victims increases. This results in people being more moved by a single child in need of help than a group of children. While studies have mainly focused on human victims, a 2013 study examined whether this applied to people’s responses to animal suffering. This was in the context of environmental threats rather than animals being directly subjected to harm by people (as in the case of animal experiments). The researchers found that the ‘compassion fade’ phenomenon did persist, but only amongst people who did not consider themselves to be environmentalists [31]. Given the millions of animals used in experiments, it is possible that the ‘compassion fade’ phenomenon could reduce public sympathy for them. Nonetheless, as discussed above, it seems that any sense of ‘compassion fade’ in relation to individual animals does not prevent the public from supporting more decisive action from policymakers in replacing animal research.

## 6. Modernizing Medical Research within the British Policy Context

Modernizing medical research would appear to support the British Government’s current policy priorities. The Government frequently mentions its ambition to make Britain a ‘science superpower’, and this has become a regular element of its post-Brexit phraseology. During the budget announcement in October 2021, for example, Chancellor Rishi Sunak stated that ‘we are making this country a science and technology superpower’ [32]. While the Government has repeatedly emphasized its commitment to providing financial support for R&D, the post-pandemic environment has seen some cooling of these commitments. The Autumn 2021 Budget and Spending Review, for example, confirmed the commitment to invest £22 billion in R&D, but this would be by 2026/27 and not the original target date of 2024/25 [32,33].

The past year has also seen some policy progress in the field of animal welfare. The 2021 Queen’s speech mentioned a commitment to high standards of animal welfare, and a number of draft Bills have followed on from this [34]. For example, the Animal Welfare (Kept Animals) Bill, which is currently going through Parliament, contains measures to tackle puppy smuggling, ban the export of live animals for fattening and slaughter and end the keeping of primates as pets [35]. Arguably one of the most significant developments has been the Animal Welfare (Sentience) Bill, which ensures that this concept is enshrined in British law post-Brexit [36]. While the bill has many shortcomings, such as its limited mechanisms for ensuring that animal sentience is adequately considered by policymakers, its existence is undoubtedly a positive step for animal protection. A survey by NFP Synergy found animals were the third most popular cause amongst MPs. This was chosen as a favorite cause by 38% of MPs (behind cancer and children and young people). In fact, the survey found that animal welfare was the top favorite cause amongst Conservative MPs (selected by 45%) [37]. The context of a Conservative government in Britain makes this especially interesting.

However, animals in laboratories have not benefitted from the recent focus on animal welfare. For example, the Government’s May 2021 Action Plan for Animal Welfare contained just one sentence about animals used in experiments, and this did not offer any new measures [38]. Clearly, the reasons for this lack of progress will be complex and multi-faceted but it is possible that the departmental allocation of different issues plays a role. Animal welfare is the responsibility of the Department for Environment, Food and Rural Affairs (Defra), which also deals with wildlife and farmed animals, while the Home Office is responsible for animals in laboratories. The minister with direct responsibility for this area has a diverse portfolio that includes countering extremism and forensic science [39] and has not shown any detailed engagement on the issue of animal experimentation.

Another key barrier to progress is regulatory. International regulations on the development of medicines, such as those produced by the International Council for Harmonization of Technical Requirements for Pharmaceuticals for Human Use (ICH), generate the clear expectation that animals should be used in tests [40]. This can contribute to a misconception amongst the public and policymakers that rigid legal requirements preclude any policy action to accelerate the replacement of animal experiments with more modern techniques. However, it is worth noting that these regulations take the form of guidelines rather than legislation and offer some flexibility for individual regulators to adapt their approach. Britain’s Medicines and Healthcare products Regulatory Agency (MHRA) has given one example where it did not require data from animal tests before a new therapy could proceed to clinical trials. While this case was unusual, it does demonstrate that some divergence from usual practices is possible [41]. Despite this apparent flexibility, it is worth noting that regulatory requirements are still an important driver of animal tests and represent a hurdle which must be addressed.

In addition, the development of the COVID-19 vaccines saw some departure from the usual linear process in which animal tests had to be completed before human clinical trials. It was agreed by the International Coalition of Medicines Regulatory Authorities (ICMRA) that ‘it is not required to demonstrate the efficacy of the SARS-CoV-2 vaccine candidate in animal challenge models prior to proceeding to [first in human] FIH clinical trials’ [42]. In March 2020, a COVID-19 vaccine from Moderna Therapeutics was trialed in humans at the same time as animal tests were carried out [43]. Whilst this does not mean that animal tests were ‘skipped’, it does demonstrate some flexibility from regulators.

Fundamentally, it should be noted that tests to satisfy regulatory requirements account for a relatively small proportion of the overall number of experiments. In 2020, only 33% of experimental ‘procedures’ in Great Britain were carried out to satisfy regulatory requirements. The largest proportion of experiments came under the category of basic research (53%), for which there is no regulatory requirement for animal use [44].

Taking decisive action to modernize medical research could help the British Government combine its ambitions to make Britain a science superpower and a world leader on animal welfare.

## 7. Modernizing Medical Research within a Global Context

Recent years have seen significant global progress in replacing animals with human relevant techniques.

One of the best-known examples of this is activity by the US Food and Drug Administration (FDA). The agency has produced a series of reports and roadmaps on the subject of replacing animals in the field of toxicity testing. These are listed in Table 1 below.

The Netherlands has also seen strong progress in this field, although the strategy has become less specific and time-bound following a change in administration. In 2016, the country’s Agriculture Minister requested that the Netherlands should become a world leader in animal-free innovations, and tasked the Netherlands National Committee for the protection of animals used for scientific purposes (NCad), with developing a phase-out plan. The work undertaken by the committee suggested that some areas of animal use for regulatory testing could be phased out as early as 2025 [48]. When the Rutte III administration took office, the Dutch Government moved away from the mention of specific dates but worked to facilitate collaboration by establishing the Transition Program for Innovation without the use of animals (TPI). This is a partnership between government, industry, academia and third sector stakeholders [49].

The TPI has spearheaded valuable collaborative initiatives such as ‘helpathons’, which won the Lush Prize for training in 2020 [50]. These collaborative workshops bring different parties together to find new ways to address research questions without the use of animals. Another important initiative is the Beyond Animal Testing Index (BATI), which aims to benchmark public research organizations according to their progress in the field of replacing animal experiments. The initiative looks at different ‘domains’ such as capacity building and research and development. The system then measures institutions’ actions according to three pillars (commitments, transparency and performance) [51].

Recent months have seen further global progress in replacing the use of animals. The Swedish Government, for example, has tasked its 3R Center with looking at what further measures can be taken to minimize animal experiments. This follows requests from animal protection groups such as Djurens Rätt to develop a plan for phasing out the use of animals [52]. In September 2021, the European Parliament voted in favor of a resolution which called for an action plan to phase out animal experiments [53]. This growing momentum around a phase-out is also reflected in Britain. Over 100,000 people have signed a petition calling on the British Government to develop a robust plan for phasing out animal experiments, and this was discussed during a debate in the UK Parliament on 25 October 2021 [54]. The implementation of a phase-out plan would also allow time for animal-free methods, such as computer modelling, to be optimized and to ensure that they were as robust as possible.

## 8. Call for a British Government Minister to Focus on Modernizing Medical Research

The transition to modern, animal-free medical research will require a number of different inputs, from large-scale dedicated funding to awareness-raising initiatives amongst scientists, to education and training. However, the importance of political will to change this paradigm cannot be overstated. Policy action has the capacity to drive and accelerate technological development and changes in practice. This principle is noted by Charlotte E. Blattner when explaining how the EU Cosmetics Directive drove significant innovation in this area [55]. The Directive banned marketing in the EU of cosmetics that had been tested on animals or contained animal-tested ingredients, as well as banning the practice of testing cosmetics on animals [56]. A 2021 article by Barthe et al. outlines the many in vitro and ex vivo methods that have been developed for the safety testing of cosmetics, including 3D models using human skin equivalent [57]. It should, of course, be noted that the Cosmetics Directive is currently being undermined by the requirements for animal tests under chemicals legislation, even for substances only used in cosmetics. However, this does not alter the technological progress that has come about as a result of the Directive. It is also worth noting the recent decision from the European Court of Justice which upheld that under REACH, animal testing must only be used as a last resort [58].

As discussed above, the implementation of concrete plans and targets by policymakers will be vital to accelerating the replacement of animals. However, the success of these will depend on policymakers’ commitment and interest in implementing them. In 2015, Innovate UK and a number of other UK bodies published *A non-animal technologies roadmap for the UK*. This contained a number of constructive recommendations, such as fostering collaboration between academia and industry, promoting the UK NAMs industry internationally to drive economic growth, and establishing an advisory board to drive implementation of the Roadmap [59]. Despite these recommendations, we are not aware that they have translated into any concrete policy action. This illustrates the pitfalls that progressive plans and policy documents can face if there is no accountability for their implementation and progress.

A measure that would drive concrete progress in this area would be establishing a new ministerial position within the British Government to accelerate the transition to human relevant science. This new Minister for Human Relevant Science could undertake key activities needed to drive change such as spearheading a declaration about the need to modernize medical research, overseeing the production of a roadmap and workplans setting out how the transition to animal-free science will take place, and taking responsibility for ensuring that these are successfully implemented. A priority area of focus would be ensuring cooperation between industry and regulators and helping to coordinate international collaboration on areas such as updating pharmaceutical regulations to facilitate the use of non-animal methods. Above all, they would act as a champion for animal-free science.

The British Government has already created ministerial roles to ensure progress on key priorities. Nadhim Zahawi, for example, was appointed as the minister for COVID-19 vaccine delivery. While the success of the program depended on vast, multi-faceted cooperation, establishing a dedicated ministerial role helped to highlight this as a key priority for the British Government. Similarly, it has been reported that the Government is considering establishing a new ministerial position to tackle the waiting lists for NHS medical treatment caused by the COVID-19 pandemic. Again, this would signal clear government prioritization and focus [60]. Appointing a Minister for Human Relevant Science would send a clear signal that Britain is serious about modernizing medical research for the benefit of animals and people. It is also worth noting the plans that are currently underway to appoint a Patient Safety Commissioner, following a recommendation from the Independent Medicines and Medical Devices Safety Review [61]. While modernizing medical research would bring obvious benefits for patient safety, appointing a dedicated Commissioner for Human Relevant Science could also help to accelerate this vital transition.

## 9. Conclusions

There is a powerful case for modernizing medical research by replacing animals with human relevant techniques. This approach will provide the best possible chance of accelerating progress in tackling the major public health challenges that have a terrible impact on patients and place strain on public healthcare systems, providing better results for people, more quickly. This transformation would help to spare animals from being subjected to experiments that inherently involve suffering. Engaging with the NAMs sector also provides significant economic opportunities. Within a British context, transforming research in this way would align with the public and policymakers’ priorities and help to place Britain at the forefront of the global shift towards animal-free research. The evidence in favor of modernizing medical research is becoming increasingly difficult to ignore, and policymakers should embrace this more ethical and future-focused approach to research.

## Figures and Tables

**Table 1 animals-12-01173-t001:** Relevant publications by the FDA.

Year	Publication	Significance
2011	Advancing Regulatory Science at FDA [45]	Identifies the need to modernize toxicology as one of eight priority areas. Includes commitment to develop tools that better predict patient responses, and to using computational methods. However, also commits to developing new ‘animal models’.
2017	FDA’s Predictive Toxicology Roadmap [46]	Sets out a series of steps to promote the development of NAMs and integrate these into regulatory assessments. These include training and improved communication with drug developers.
2021	Advancing New Alternative Methodologies at FDA [47]	Developed by the Alternative Methods Working Group, reports on progress in NAM development and integration across the FDA.

## Data Availability

Not applicable.

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
