# Peer review of "Modernizing Medical Research to Benefit People and Animals"

_animals, 2022, doi:10.3390/ani12091173_

Round 1
Reviewer 1 Report
This article describes a relevant and timely topic. In the abstract the scientific, ethical and economic arguments are mentioned; as this publication is for a special issue also focussing on the regulatory perspective of current efforts towards Replacement, please add that topic to the abstract. It is also important to stress the international complexity in e.g. regulatory demands, next to Britain alone.
Line 114 mentions force feeding: this is not only an animal suffering issue, it also has scientific implications. Force feeding causes stress and slows down gastrointestinal function and will thus influence results.
Line 156: add references from e.g. Thomas Hartung or the Joint Research Center to make it more general.
Line 162: you can refer to Judith van Luijk's work about questionnaires in the scientific community on 3R / non animal methods.
Line 173: EU e-learning module 52 on how to search for existing non animal methods, which has been developed recently, freely available on the ETPLAS learning platform. Suggest to add this reference and advise to include in education.
Line 181: suggest to add 'compassion fatigue'
Line 201-209: Refer to Deloitte LLP 2021 on the low return on investment for drugs. In 2019 it was only 1.6%, in 2020 it was 2.5%. Pharma is in the danger zone in the current system.
line 210-222: add the estimated costs of research waste (Lancet series 2014) and how little of total funding is given to non-animal methods (estimates are less than 1%).
line 261-263: I suggest rephrasing, sentence not clear.
paragraph 5 (line 223): were there any indications of a change of views on the use of animals as a result of the Covid pandemic? Were animal studies considered more essential and/or were alternatives considered better?
Line 304 and further: even thought there is some flexibility among regulators, still many animal studies are being done to satisfy regulatory requirements. This is considered a hurdle and needs further research to examine this in-depth why it is there and how that could be changed.
Line 318-320: add reference (link).
Line 382: suggest to add the ESSO court case against ECHA, that 'animals are only to be used as a last resort.' (Fentem et al. ATLA 2021)
Line 385-405: wouldn't a cooperation between industry and regulators need to be the first priority to achieve actual progress?
Paragraph 9 Conclusion: add economic advantages. Alternatives are expected to lead to faster and better results.
Reviewer 2 Report
This manuscript is a well-written and nicely organized commentary presenting the justifications of the authors for ceasing the use of animals in research in the UK.
Many of the web-based reference links did not work properly for me so were not usable or useful. I suggest rechecking all of these. Many of these were not academic, data-driven sources and likely subject to change frequently so may become obsolete quickly once repaired. This may limit their usefulness as references for a published manuscript and should be given thoughtful consideration by authors and editors before use.
Additional information that may strengthen the manuscript include responses to alternative views, such as the following:
- Section 3 argues the ethical case for modernising medical research. Did the authors consider the ethics of initial trials of novel treatments such as gene therapy? Is it ethical to pilot these in humans? Or is this a case that is ethical to pilot in animals first for mechanism and safety before use in humans? Manipulations to germ cells, with CRISPR, for instance?
- Section 4 argues the economic case for cost savings, lowering the cost of new medicines, and speeding drug discovery. However, ending animal research will not guarantee that medical discoveries occur at a more rapid pace, and some computer models are based on biological disease data collected from animals and humans. Is there a case to be made for a phase out of animal research to ensure that work in progress and computer modeling being developed is completed and robust?
- Section 5 suggests a binding plan for ending animal testing (lines 232-233). What about animal research for animals' sakes? Heartworm preventative, deworming medications, antibiotics, analgesics, extended duration formulations of analgesics? Are these not allowed for testing in animals? Or are the opinions in this manuscript strictly in the context of using animals for research for the benefit of humans?
- Section 8 discusses patient safety (lines 414-418). Did the authors consider alternatives to medical device prototype refinement, testing, and development to assure function and safety before implantation in humans? Reference # 53 is not working for me, so I am unable to check the content of this reference. Please accept my apology if the answer is contained in that reference.
- Reference 7 was used on lines 85-88 as an example of computer-based analysis used to examine changes in gene expression, and by existing drugs, to allow repurposing of drugs. However, the authors did not mention that the computer analysis was carefully partnered with targeted animal experiments to demonstrate efficacy and safety before embarking on human trials. Again, I am curious as to whether a thoughtful middle ground may exist or if one extreme or the other regarding animal research must be the answer.
